# WaveGS: Physics-Inspired Wavelet Splatting for Thermal Novel View Synthesis

## Abstract

3D thermal infrared reconstruction aims to reconstruct a three-dimensional model with thermal distribution information from multi-view thermal images or video sequences. Recent studies have shown that incorporating thermodynamic knowledge into 3D representations can achieve superior novel view synthesis performance. However, without data capturing the temporal evolution or providing temperature calibration, the representation learning becomes ill-posed. To address this problem, our key insight is to leverage the low-pass characteristic of heat conduction to model scene representations in the frequency domain. To this end, we first represent the 3D thermal field using a continuous Vector-Matrix (VM) decomposition, and parameterize the resulting factors with a learnable wavelet basis. This allows us to explicitly disentangle the scene representation into low-frequency components that capture smooth thermal variations and high-frequency subbands that encode structural details. Next, we devise a high-frequency masking strategy to suppress infrared noise while preserving salient details. Concurrently, this mask guides a learnable geometric deformation field to optimize geometric details by directly adjusting the anchor positions, thereby eliminating the need for explicit material parameters. Finally, the modulated wavelet coefficients are dynamically reconstructed into a spatial-domain feature field via a differentiable inverse wavelet transform. Extensive experiments on four datasets demonstrate that WaveGS consistently outperforms existing methods across multiple metrics.

## 1 Introduction

Thermal imaging technology captures the thermal radiation of objects using sensors, converting temperature information into visual images. Infrared thermography offers all-weather imaging capabilities, unconstrained by illumination or weather conditions, and has found widespread application in fields such as pedestrian recognition Shi et al. (2024b; 2023; 2024a); Lin et al. (2024), medicine Ma et al. (2023); Ring & Ammer (2012), industrial inspection Chen et al. (2024a); Wang et al. (2025a); Zhang et al. (2025a), and agriculture Gao et al. (2023); Wang et al. (2025b). The principles of heat conduction cause thermal energy to diffuse across object surfaces, resulting in images that are inherently smooth and lack distinct textural features. Consequently, synthesizing novel views from static thermal images without temporal evolution or temperature calibration becomes a severely ill-posed problem. Specifically, the same observation can correspond to multiple volumetric density distributions, and the absence of distinct textural features makes the precise recovery of geometric structures challenging.

Traditional 3D thermal reconstruction Acampora et al. (2011); Cao et al. (2018); Kriczky et al. (2015); Abreu de Souza et al. (2023) commonly incorporates RGB imagery for geometric guidance, as thermal images inherently suffer from the "ghosting effect" that obscures geometric textures. This phenomenon arises from the superposition of object-emitted thermal radiation and ambient-reflected radiation, leading to distorted temperature measurements and loss of surface texture details. Although recent neural rendering approaches such as NeRF Mildenhall et al. (2021) and 3D Gaussian Splatting (3DGS) Kerbl et al. (2023) demonstrate capabilities in novel view synthesis from thermal infrared data, physical constraint integration remains challenging. NTR-Gaussian Yang et al. (2025) attempts to address this through inversion of thermodynamic parameters (e.g., emissivity $\epsilon$ and convective heat transfer coefficient $h$), but this constitutes an ill-posed inverse problem. The absence of ground-truth thermodynamic labels, combined with the non-uniqueness of parameter so-

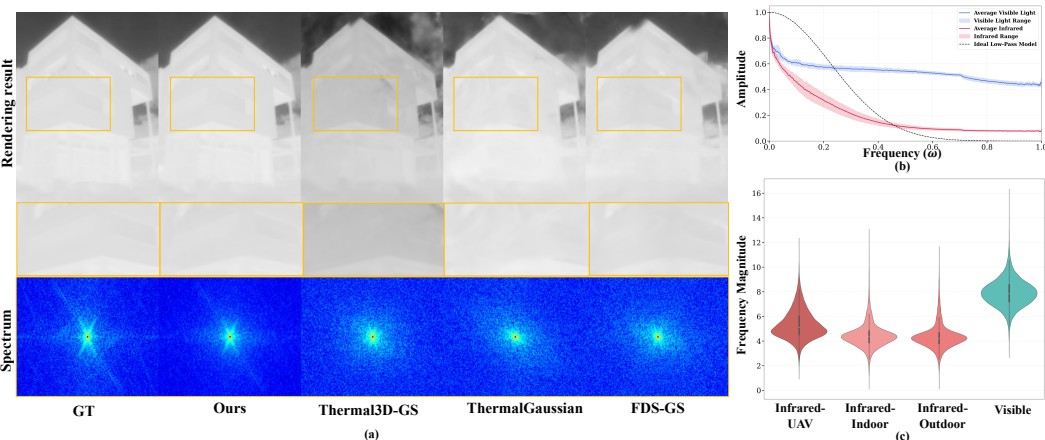

Figure 1: **(a)** Our method faithfully reproduces the low-frequency spectrum of GT and avoids the structural blurring that occurs in other methods. **(b) Statistical Distribution of Frequency Magnitudes:** Compared to visible light images, the signal energy in infrared images is predominantly contributed by low-frequency components. **(c) Spectral Decay Curve:** The amplitude of infrared images attenuates rapidly with increasing frequency, and its attenuation trend closely matches the characteristics of an ideal low-pass filter model $e^{-k\|\boldsymbol{\omega}\|^2}$.

lutions producing similar thermal observations, forces researchers to adopt overly simplified energy transfer assumptions. Such limitations hinder the accurate modeling of complex real-world thermal processes.

We identify two fundamental challenges inherent to novel view synthesis in thermal imaging. Thermal infrared imagery intrinsically lacks the rich textures and distinct edge details characteristic of visible-light images. As illustrated in Figure 1(b), the signal energy in thermal images is predominantly concentrated in the low-frequency bands, posing a significant limitation for existing reconstruction methods that rely on spatial domain feature extraction. Secondly, the amplitude of thermal signals rapidly attenuates with increasing frequency, a trend that closely matches an ideal low-pass filter model, as shown in Figure 1(c). It indicates that the low-pass filtering behavior intrinsic to heat conduction causes thermal data to exhibit pronounced spectral energy concentration at lower frequencies. This intrinsic physical property provides a powerful and natural inductive bias, prompting us to utilize a low-pass prior to guide and constrain the reconstruction process.

To address the aforementioned challenges, we propose **WaveGS**, a framework grounded in physical priors that employs a wavelet decomposition of the 3D feature field. By parameterizing the scene's 3D thermal features in the frequency domain, our approach directly embeds the inherent low-pass characteristic of heat conduction into the learning process. Following Scaffold-GS Lu et al. (2024b), our representation of the thermal field leverages a continuous Vector-Matrix (VM) decomposition Sun & Ansari (2016); Chen et al. (2022). This decomposition is based on a sparse set of anchor points, a strategy that facilitates direct frequency-domain modeling and obviates the use of dense voxel grids. Each vector and matrix factor is parameterized by learnable wavelet bases, resulting in an explicitly frequency-aware 3D representation. Through multiscale wavelet decomposition, the feature field is disentangled into low-frequency components that capture smooth thermal distributions and high-frequency subbands that encode structural details. This design aligns with the physical nature of heat conduction and allows efficient modulation via frequency-guided priors. We introduce a differentiable mask to enforce structural sparsity by isolating salient high-frequency components. This mask preserves critical details while mitigating noise, and is complemented by an $L_1$ regularization loss that promotes a compact representation. For geometric fidelity, we employ the high-frequency mask to guide anchor deformation through learned offsets, constrained by regularization losses enforcing spatial locality and local smoothness. The modulated wavelet coefficients are then reconstructed into the spatial domain via differentiable inverse wavelet transform. Extensive experiments on four datasets demonstrate the effectiveness of our framework. We summarize our main contributions as follows:

- We introduce WaveGS, the first framework to parameterize 3D thermal radiation scenes in the wavelet domain. By leveraging the wavelet transform, we directly embed the inherent low-pass physical characteristic of heat conduction into the 3D scene representation.

- We propose a geometric deformation field guided by a sparsity-inducing high-frequency mask, where the field performs local geometric corrections to achieve a precise reconstruction of key details.

- WaveGS markedly enhances the visual quality of thermal novei view synthesis, demonstrating competitive performance on both benchmarks in comparison to state-of-the-art methods, and facilitating real-time rendering speeds exceeding 200 FPS.

## 2 RELATED WORK

### 2.1 THERMAL 3D RECONSTRUCTION

3D thermal reconstruction aims to generate a three-dimensional representation of a scene with thermal information from multi-view thermal infrared imagery. Early works Schramm et al. (2022) addressed the limited spatial resolution of thermal images by incorporating auxiliary modalities. Rangel et al. Rangel et al. (2014) fused thermal and depth data via multimodal calibration, while Zhao et al. Zhao et al. (2017) developed a real-time SLAM system Taketomi et al. (2017) combining thermal and RGB-D inputs through depth-based ICP alignment. Li et al. Li et al. (2023) enhanced thermal stereo matching with denoising, tailored calibration, and weighted SGBM. AT-Loc Liu et al. (2024a) extended thermal reconstruction to aerial localization using a dedicated dataset and a geometry-aware render-to-localization pipeline. Recent efforts increasingly adopt learning-based methods. ThermoNeRF Hassan et al. (2024) extends Neural Radiance Fields to disentangle color and temperature via cross-modal supervision. Thermal3D-GS Chen et al. (2024b) introduces physics-informed Gaussian Splatting with atmospheric and conduction constraints. ThermalGaussian Lu et al. (2024a) further integrates cross-modal calibration and physically motivated smoothing. NTR-Gaussian Yang et al. (2025) models thermal radiation with neural fields to forecast dynamic temperature evolution, under nighttime conditions.

### 2.2 FREQUENCY ANALYSIS

Frequency analysis Zadeh (1950) is a foundational technique in signal processing that decomposes signals or images into components across different frequencies and scales. In visual computing, it supports detail preservation, denoising, and multi-scale modeling. The Discrete Wavelet Transform (DWT) Farge et al. (1992); Zhang (2019), in particular, provides excellent spatial-frequency localization, making it effective for capturing fine-grained high-frequency variations while preserving global structure. In neural scene representation Zhao et al. (2025); Li et al. (2024); Liu et al. (2024b); Xie et al. (2024), frequency-domain modeling has been increasingly adopted to improve rendering fidelity and generalization. Zhang et al. Zhang et al. (2025b) emphasize the role of frequency-aware operations in NeRF architectures for complex scenes. WaveNeRF Xu et al. (2023) introduces wavelet-based frequency volumes to extract geometry-aware features for generalizable novel view synthesis. TriNeRFLet Khatib & Giryes (2024) builds on this by learning wavelet-based triplane representations, using coarse coefficients from low-resolution inputs and refining them via multi-scale supervision with pre-trained guidance. These ideas have recently extended to 3D Gaussian Splatting Zuo et al. (2025). FreGS Zhang et al. (2024) applies Fourier transforms to rendered and ground-truth images, enforcing frequency-domain consistency to mitigate aliasing. FDS-GS Zeng et al. (2025) reparameterizes Gaussian scaling to explicitly link spatial scale with frequency attenuation, enhancing consistency in spectral behavior.

## 3 PROPOSED METHOD

### 3.1 OVERVIEW

Given a set of multi-view thermal infrared images, our objective is to synthesize novel views of the target scene. Inspired by the frequency-domain solution of heat conduction, our method reconstructs the thermal scene by explicitly modeling the low-pass characteristics inherent in heat transfer.

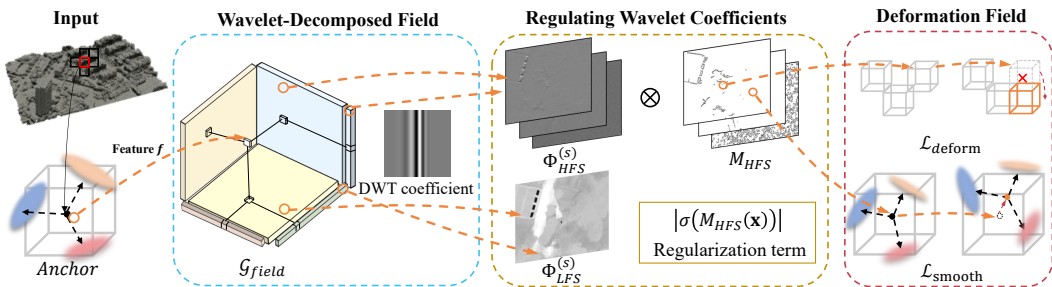

Figure 2: **Overview of the proposed pipeline.** Our method takes as input a 3D scene represented by a sparse set of anchor points. To model the scene's feature field, we adopt a wavelet-decomposed representation $\mathcal{G}_{\text{field}}$ that separates the field into low-frequency coefficients $\Phi_{LFS}$, which capture smooth variations, and high-frequency coefficients $\Phi_{HFS}$, which encode structural details. The high-frequency coefficients are further refined by a learnable mask $M_{HFS}$, which selectively preserves salient structures and promotes compactness through regularization. This mask then guides a learnable deformation field that adjusts the anchor positions, subject to a deformation loss $\mathcal{L}_{\text{deform}}$ and a smoothness loss $\mathcal{L}_{\text{smooth}}$.

We achieve this by employing wavelet decomposition to represent scene features in the frequency domain. An overview of the proposed model is illustrated in Figure 2.

### 3.2 PRELIMINARY

**Scaffold-GS** Lu et al. (2024b) is a neural scene representation framework that extends 3D Gaussian Splatting by introducing a learnable scaffold structure to enhance geometric fidelity and rendering quality. Instead of representing scenes with millions of independent Gaussians, Scaffold-GS defines each neural Gaussian's position $\boldsymbol{\mu}_{ij}$ relative to a scaffold anchor point $\mathbf{x}_{v_i}$ using a learnable scale $\mathbf{l}_{v_i}$ and an offset vector $\mathbf{o}_j$:

$$\boldsymbol{\mu}_{ij} = \mathbf{x}_{v_i} + \mathbf{l}_{v_i} \cdot \mathbf{o}_j \tag{1}$$

To predict Gaussian attributes, an MLP takes the anchor's local feature vector $\mathbf{f}_{v_i}$ and the canonical viewing direction $\mathbf{d}_{v_c}$ as input, generating the opacity $\alpha_j$, color $\mathbf{c}_j$, rotation $\mathbf{q}_j$ and scale $\mathbf{s}_j$:

$$\{\alpha_j, \mathbf{c}_j, \mathbf{q}_j, \mathbf{s}_j\} = \text{MLP}(\mathbf{f}_{v_i}, \mathbf{d}_{v_c}) \tag{2}$$

**Heat Conduction.** The heat equation Widder (1976); Necati et al. (1993); Bao et al. (2023); Wang et al. (2025c) in $\mathbb{R}^n$ for a temperature field $u(\mathbf{x}, t)$ with thermal diffusivity $k > 0$ is:

$$\frac{\partial u}{\partial t} = k\nabla^2 u, \quad u(\mathbf{x}, 0) = f(\mathbf{x}) \tag{3}$$

Let $\overline{u}(\boldsymbol{\omega}, t) = \mathcal{F}(u(\mathbf{x}, t))$ and $\overline{f}(\boldsymbol{\omega}) = \mathcal{F}(f(\mathbf{x}))$. Applying the Fourier Transform Narasimhan (1999) converts the PDE into an ODE using the properties $\mathcal{F}(\partial_t u) = \partial_t \overline{u}$ and $\mathcal{F}(\nabla^2 u) = -\|\boldsymbol{\omega}\|^2 \overline{u}$:

$$\frac{d\overline{u}}{dt} = -k\|\boldsymbol{\omega}\|^2 \overline{u} \tag{4}$$

The solution in the frequency domain, with the transformed initial condition, is:

$$\overline{u}(\boldsymbol{\omega}, t) = \overline{f}(\boldsymbol{\omega})e^{-k\|\boldsymbol{\omega}\|^2 t} \tag{5}$$

The term $e^{-k\|\boldsymbol{\omega}\|^2 t}$ acts as a low-pass filter. The high frequency components decay rapidly with time $t$, while the low frequency components persist. This inherent smoothing aligns with physical intuition: Heat diffusion suppresses sharp thermal gradients and preserves temperature distributions.

### 3.3 WAVELET-DECOMPOSED 3D THERMAL FIELD

In thermal scene modeling, the inherent low-pass nature of heat diffusion smooths high-frequency spatial details. A key limitation of current methods is their dependence on auxiliary voxel grids Sun

et al. (2022); Zhang et al. (2023) for direct frequency-domain optimization of 3DGS. These grids are a known source of both significant computational overhead and interpolation inaccuracies. To overcome this, we adopt an anchor-based approach inspired by Scaffold-GS, enabling direct frequency modeling on a set of sparse anchors without a grid. We represent the 3D scene as a continuous feature field, $\mathcal{G}$, constructed using a Vector-Matrix (VM) decomposition over these anchors. For any spatial coordinate $\mathbf{x} = (x_1, x_2, x_3)$, the feature $\mathcal{G}_{\text{field}}(\mathbf{x})$ is queried as:

$$\mathcal{G}_{\text{field}}(\mathbf{x}) = \sum_{r=1}^{R} \sum_{i=1}^{3} \mathbf{v}_{r,i}(x_i) \odot \mathbf{M}_{r,i}(x_j, x_k) \tag{6}$$

Here, for each $i \in \{1, 2, 3\}$, the pair $(x_j, x_k)$ denotes the remaining two spatial dimensions orthogonal to $x_i$. The vector $\mathbf{v}_{r,i}(\cdot)$ captures the directional variation along axis $x_i$, while the matrix $\mathbf{M}_{r,i}(\cdot, \cdot)$ encodes context-dependent correlations in the complementary 2D plane. This decomposition provides an efficient low-rank factorization of the 3D feature field.

Directly learning the VM decomposition factors poses two major challenges: the difficulty in distinguishing smooth low-frequency thermal distributions from sharp high-frequency geometric edges and a high susceptibility to thermal noise. To address these issues, we employ wavelet transforms Chun-Lin (2010), which offer excellent spatial-frequency localization. This enables the signal to be decomposed into low-frequency components, which we denote $\Phi_{LFS}$, capturing smooth thermal variations, and high-frequency subbands, collectively denoted $\Phi_{HFS}$, which encode structural details. Such decomposition facilitates the incorporation of physical priors, allowing for more effective encoding of both the feature field and anchor attributes.

Leveraging the multi-scale nature of wavelet decomposition, we construct a hierarchical feature field. For the one-dimensional vector factors $\mathbf{v}_r$, we employ a multi-scale wavelet decomposition:

$$\mathbf{v}_r(x) = \sum_k \Phi_k^{(0)} \phi_k(x) + \sum_{s=1}^{S} \sum_k \Phi_k^{(s)} \psi_k^{(s)}(x), \tag{7}$$

where $\phi_k(x)$ denotes the scaling functions forming the low-frequency basis, $\psi_k^{(s)}(x)$ are the wavelet functions at scale $s$, and $\Phi_k^{(0)}$, $\Phi_k^{(s)}$ represent the corresponding learnable coefficients.

In parallel, we extend this multi-scale wavelet modeling to the two-dimensional matrix factors $\mathbf{M}_r$. A standard 2D wavelet transform decomposes a signal into one low-frequency approximation subband (LL) and three high-frequency detail sub-bands that capture horizontal (LH), vertical (HL), and diagonal (HH) features. Accordingly, we represent each matrix factor $\mathbf{M}_{r,i}(x_j, x_k)$ as a complete linear combination of these 2D wavelet bases:

$$\begin{aligned}
\mathbf{M}_{r,i}(x_j, x_k) = &\sum_{m,n} \Psi_{LL,m,n}^{(0)} \phi_m(x_j)\phi_n(x_k) \\
&+ \sum_{s=1}^{S} \sum_{m,n} \Big( \Psi_{LH,m,n}^{(s)} \phi_m(x_j)\psi_n^{(s)}(x_k) \\
&\qquad + \Psi_{HL,m,n}^{(s)} \psi_m^{(s)}(x_j)\phi_n(x_k) \\
&\qquad + \Psi_{HH,m,n}^{(s)} \psi_m^{(s)}(x_j)\psi_n^{(s)}(x_k) \Big)
\end{aligned} \tag{8}$$

Here, the representation is parameterized by distinct sets of learnable coefficients: $\Psi_{LL,m,n}^{(0)}$ for the base approximation, and $\Psi_{LH,m,n}^{(s)}$, $\Psi_{HL,m,n}^{(s)}$, and $\Psi_{HH,m,n}^{(s)}$ for the horizontal, vertical, and diagonal details at each scale $s$, respectively. This complete formulation ensures that our feature field is explicitly parameterized in the frequency domain, enabling a physically-informed representation where low-pass thermal propagation and sharp structural variations across different orientations are separately captured and regulated.

## 3.4 REGULATING WAVELET COEFFICIENTS FOR PHYSICAL PRIORS

To enhance the physical plausibility and structural accuracy of the reconstructed thermal fields, we introduce a targeted modulation scheme on the decoupled wavelet coefficients. This scheme is designed to promote geometric sparsity via high-frequency masking.

**Sparsity-Inducing High-Frequency Masking.** To encode structural priors and promote sparse activation across the high-frequency sub-bands, we introduce a learnable masking mechanism. This mechanism operates on the three distinct detail components (LH, HL, HH) to generate a unified structural mask. First, we compute an energy map for each of the three high-frequency components at each scale $s$:

$$M_b^{(s)}(\mathbf{x}) = \left\| \Phi_b^{(s)}(\mathbf{x}) \right\|_2, \quad \text{for } b \in \{\text{LH, HL, HH}\} \tag{9}$$

These per-band energy maps are then aggregated into a single high-frequency energy map for each scale by summation:

$$M_{HFS}^{(s)}(\mathbf{x}) = M_{LH}^{(s)}(\mathbf{x}) + M_{HL}^{(s)}(\mathbf{x}) + M_{HH}^{(s)}(\mathbf{x}) \tag{10}$$

We then aggregate these multi-scale responses using a weighted summation to produce a fused mask:

$$M_{HFS}^{\text{fused}}(\mathbf{x}) = \sum_{s=1}^{S} w_s \cdot \sigma \left( M_{HFS}^{(s)}(\mathbf{x}) \right), \tag{11}$$

where $w_s$ are learnable weights and $\sigma(\cdot)$ denotes the sigmoid function. This unified mask is then applied to all three high-frequency sub-bands using a straight-through estimator (STE) to enforce sparsity while maintaining differentiability:

$$\hat{\Phi}_b = \text{sg} \left( \left( \mathcal{H}(M_{HFS}^{\text{fused}}) - \sigma(M_{HFS}^{\text{fused}}) \right) \odot \Phi_b \right) \\ + \sigma(M_{HFS}^{\text{fused}}) \odot \Phi_b, \quad \text{for } b \in \{\text{LH, HL, HH}\} \tag{12}$$

where $\mathcal{H}(\cdot)$ is the Heaviside step function and $\text{sg}(\cdot)$ denotes the stop-gradient operator. We further regularize the fused mask via an $L_1$ penalty to promote a compact representation:

$$\mathcal{L}_{\text{sparsity}} = \lambda_s \cdot \sum_{\mathbf{x}} \left| \sigma \left( M_{HFS}^{\text{fused}}(\mathbf{x}) \right) \right|. \tag{13}$$

In each forward pass, the spatial-domain factors $\mathbf{v}_r$ and $\mathbf{M}_r$ are reconstructed from the masked high-frequency wavelet coefficients and the low-frequency coefficients via an inverse wavelet transform. The final anchor features are then aggregated using the tensor decomposition formulation.

## 3.5 WAVELET-GUIDED GEOMETRIC DEFORMATION

In thermal imagery, non-thermal sources like emissivity discontinuities introduce high-frequency artifacts that are not indicative of thermal gradients Chapman (1986). These artifacts cause positional ambiguity, degrading the accuracy and sharpness of the 3D geometric reconstruction. As directly deforming all Gaussians is computationally intractable, our method applies a learnable deformation field to the sparse set of anchor points. This field predicts a positional offset for each anchor, enabling localized corrections to the underlying geometry. Let $\mathbf{x}_a$ denote the position of an anchor point $a$. We define a learnable offset vector $\Delta(\mathbf{x}_a) \in \mathbb{R}^3$ for each anchor, yielding the updated anchor position:

$$\mathbf{x}_a^{\text{new}} = \mathbf{x}_a + \Delta(\mathbf{x}_a). \tag{14}$$

This offset enables localized spatial corrections to the geometry, particularly in structurally informative regions identified via high-frequency activations.

To ensure the deformation field only acts where needed, it should be activated exclusively in regions where the high-frequency mask has a significant response. We enforce this locality by introducing a mask-modulated regularization term on the deformation magnitude:

$$\mathcal{L}_{\text{deform}} = \sum_{a} \left( 1 - \sigma \left( M_{HFS}^{\text{fused}}(\mathbf{x}_a) \right) \right) \cdot \|\Delta(\mathbf{x}_a)\|^2, \tag{15}$$

This loss term penalizes deformations outside the high-frequency regions, forcing $\Delta(\mathbf{x}_a)$ towards zero and confining the geometric adjustments to structurally relevant areas.

Furthermore, within the activated high-frequency regions, the deformation field must remain smooth to avoid disrupting the geometric coherence of existing structural boundaries. To enforce this, we introduce a weighted local smoothness constraint:

$$\mathcal{L}_{\text{smooth}} = \sum_{a} \sum_{\mathbf{x}_b \in \mathcal{N}(\mathbf{x}_a)} \sigma \left( M_{HFS}^{\text{fused}}(\mathbf{x}_a) \right) \cdot \|\Delta(\mathbf{x}_a) - \Delta(\mathbf{x}_b)\|^2, \tag{16}$$

where $\mathcal{N}(\mathbf{x}_a)$ is the set of neighboring anchor points for anchor $a$. This loss can be interpreted as a weighted Total Variation (TV) regularization on the deformation field, promoting smoothness specifically within high-frequency areas as indicated by the mask.

Figure 3: **Qualitative Comparison on TI-NSD.** Our method reconstructs the textures of building facades, details of ground reflections, and thermal features of foreground objects with significantly higher fidelity and clarity, resulting in renderings that are most consistent with the ground truth.

| Method | RGBT-Scenes | | | ThermoScenes | | | TI-NSD | | | NTR | | |
|---|---|---|---|---|---|---|---|---|---|---|---|---|
| | PSNR↑ | SSIM↑ | LPIPS↓ | PSNR↑ | SSIM↑ | LPIPS↓ | PSNR↑ | SSIM↑ | LPIPS↓ | PSNR↑ | SSIM↑ | LPIPS↓ |
| 3DGS | 24.467 | 0.866 | 0.202 | 30.659 | 0.965 | 0.091 | 32.436 | 0.936 | 0.202 | 32.178 | 0.964 | 0.205 |
| 4DGS | 23.739 | 0.822 | 0.229 | 22.443 | 0.713 | 0.296 | 33.955 | 0.908 | 0.114 | 30.631 | 0.952 | 0.358 |
| 2DGS | 25.310 | 0.883 | 0.181 | 30.630 | 0.965 | 0.098 | 31.810 | 0.943 | 0.217 | 31.584 | 0.952 | 0.294 |
| Thermal3D-GS | 24.872 | 0.873 | 0.189 | 25.733 | 0.929 | 0.101 | 34.938 | 0.956 | 0.188 | 30.656 | 0.933 | 0.391 |
| ThermalGaussian | 25.086 | 0.876 | 0.186 | 27.269 | 0.974 | 0.133 | 32.429 | 0.891 | 0.224 | 32.061 | 0.963 | 0.234 |
| Scaffold-GS | 23.827 | 0.860 | 0.207 | 29.508 | 0.957 | 0.100 | 31.305 | 0.930 | 0.217 | 31.233 | 0.950 | 0.319 |
| Mip-Splatting | 22.832 | 0.786 | 0.249 | 30.441 | 0.961 | 0.094 | 30.283 | 0.917 | 0.223 | 31.827 | 0.965 | 0.280 |
| FDS-GS | 25.412 | 0.884 | 0.173 | 33.263 | 0.977 | 0.071 | 33.178 | 0.949 | 0.189 | 30.944 | 0.945 | 0.298 |
| NTR-Gaussian | - | - | - | - | - | - | - | - | - | 27.765 | 0.939 | 0.263 |
| **WaveGS** | 26.197 | 0.897 | 0.157 | 33.671 | 0.978 | 0.064 | 34.992 | 0.958 | 0.182 | 32.197 | 0.968 | 0.174 |

Table 1: **Quantitative Comparison on RGBT-Scenes, ThermoScenes, TI-NSD, and NTR.** Comparison of different methods for thermal infrared novel view synthesis. For each dataset, we color the cells as best , second best , and third best .

# 4 EXPERIMENTS

## 4.1 EXPERIMENT SETUP

**Datasets.** The proposed model is evaluated on four benchmark datasets: **RGBT-Scenes** Lu et al. (2024a), a multi-view dataset of over 1,000 spatially aligned and synchronously captured TIR-RGB image pairs; **TI-NSD** Chen et al. (2024b), a large-scale benchmark for TIR novel-view synthesis with over 6,600 images from 20 diverse scenarios; **ThermoScenes** Hassan et al. (2024), the benchmark for joint RGB-thermal 3D reconstruction and synthesis across 16 real-world scenes; and **NTR Dataset** Yang et al. (2025), a dynamic thermal dataset featuring four UAV-captured scenes imaged at different nighttime intervals.

**Implementation.** We implemented our method, wavegs, in the PyTorch framework, building upon the Scaffold-GS codebase. All models were trained on a single NVIDIA RTX 4090 GPU. For the wavelet decomposition, we utilized the Biorthogonal 6.8 (Bior6.8) wavelet Joseph & Sturges (1978). The low-frequency approximation coefficients (the LL sub-band) were initialized randomly, whereas the high-frequency detail coefficients (the LH, HL, and HH sub-bands) were initialized to zero. The models were trained for a total of 30,000 iterations.

**Metric.** We evaluate novel view synthesis using PSNR, SSIM, and LPIPS to measure perceptual and structural fidelity.

**Baseline.** We compare our method against a diverse set of baselines, including general 3D representations (3DGS, Scaffold-GS, 2DGS Huang et al. (2024), 4DGS Wu et al. (2024)), frequency-guided methods (Mip-Splatting Yu et al. (2024), FDS-GS), and physics-informed thermal synthesis approaches (Thermal3D-GS, ThermalGaussian).

## 4.2 QUANTITATIVE AND QUALITATIVE COMPARISON

We evaluated our method through a diverse set of experiments that included indoor and outdoor environments, scenes with high-intensity heat sources, and aerial perspectives from UAVs. In indoor and outdoor settings, our approach reconstructs the facades of buildings, the reflections of the ground and the thermal characteristics of the foreground objects with significantly higher fidelity and clarity (Figure 3). For scenes featuring high-intensity heat sources, our method successfully

| Wavelet Basis | PSNR↑ | SSIM↑ | LPIPS↓ |
|---|---|---|---|
| WaveGS(Haar) | 24.590 | 0.864 | 0.225 |
| WaveGS(Db4) | 21.157 | 0.824 | 0.293 |
| WaveGS(DMey) | 22.170 | 0.853 | 0.259 |
| WaveGS(Coif4) | 24.116 | 0.931 | 0.316 |
| WaveGS(Sym4) | 25.360 | 0.875 | 0.141 |
| WaveGS(Bior4.4) | 25.731 | 0.888 | 0.174 |
| WaveGS(Bior6.8) | 26.197 | 0.897 | 0.157 |

| | $\mathcal{L}_{sparsity}$ | $\mathcal{L}_{deform}$ | $\mathcal{L}_{smooth}$ | PSNR↑ | SSIM↑ | LPIPS↓ |
|---|---|---|---|---|---|---|
| (1) | Baseline | | | 28.238 | 0.924 | 0.242 |
| (2) | ✓ | | | 32.080 | 0.939 | 0.204 |
| (3) | ✓ | ✓ | | 32.849 | 0.950 | 0.193 |
| (4) | ✓ | | ✓ | 33.162 | 0.953 | 0.196 |
| (5) | ✓ | ✓ | ✓ | 34.992 | 0.958 | 0.182 |

Table 2: **Ablation Studies.** (Left) Quantitative results of different wavelet bases on RGBT-Scenes. (Right) Ablation on ThermoScenes with different loss components. For each metric, we color the cells as best , second best , and third best .

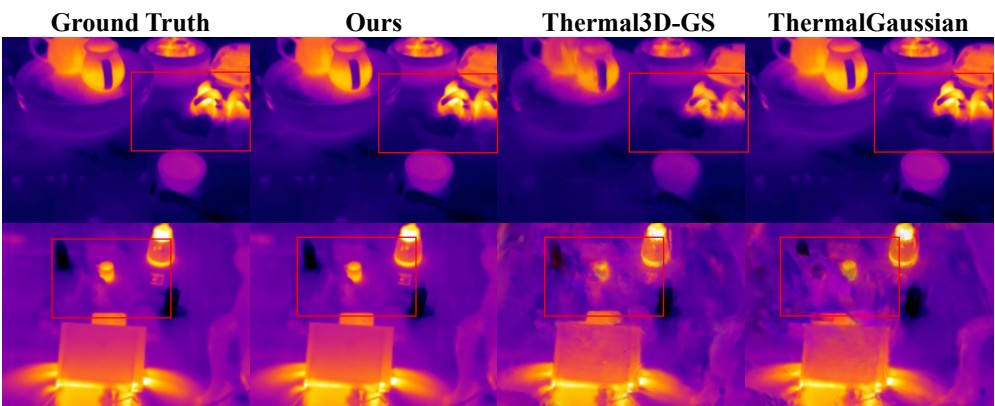

Figure 4: **Qualitative Comparison on RGBT-Scenes.** As Thermal3D-GS and Thermal Gaussian exhibit noticeable blurring or artifacts when processing high-intensity heat sources. In contrast, our method successfully preserves sharp edges and internal textures.

preserves sharp edges and internal textures, avoiding the noticeable blurring and artifacts present in competing methods like Thermal3D-GS and ThermalGaussian (Figure 4). In aerial scenes, our method generates crisp, realistic images by effectively reconstructing high-frequency details such as road contours and ground textures (Figure 5). As demonstrated in Table 1, our proposed method achieves State-of-the-Art (SOTA) performance on nearly all core metrics on the four benchmark datasets. Compared to physics-inspired thermal reconstruction methods like Thermal3D-GS and ThermalGaussian, our approach shows a comprehensive lead. For example, on the ThermoScenes dataset, our PSNR of 33.671 far exceeds the 25.733 of Thermal3D-GS and 27.269 of ThermalGaussian, indicating superior accuracy and perceptual quality. Moreover, our method outperforms the strong frequency-domain baseline FDS-GS, which leverages spectral priors for geometry and appearance modeling. Although FDS-GS ranks second on several evaluation metrics, our approach consistently achieves the best overall performance. This consistent lead highlights our method's more effective and precise utilization of frequency-domain priors, particularly in accurately recovering high-frequency structural details.

## 4.3 ABLATION STUDY

**Analysis on Wavelet Bases.** We conducted an ablation study to evaluate the influence of different wavelet bases on the performance of our thermal 3D reconstruction pipeline. The quantitative results are presented in Table 2. Among the candidates tested, the Biorthogonal 6.8 (Bior6.8) wavelet achieves the most consistent performance. It ranks first in PSNR and second in both SSIM and LPIPS. This suggests that Bior6.8 offers a good trade-off between low-frequency approximation and high-frequency detail preservation. Its strong performance may be due to its symmetric structure and balanced support width. In contrast, traditional wavelets such as Daubechies 4 (Db4) and Discrete Meyer (DMey) perform significantly worse, especially in terms of PSNR and SSIM. Interestingly, Coiflet 4 (Coif4) achieves the highest SSIM, indicating strong structural similarity.

Figure 5: **Qualitative Comparison on NTR.** In the aerial scenes, the result from Thermal3D-GS is severely blurry, losing the vast majority of details. While NTR-Gaussian and ThermalGaussian can roughly reconstruct the scene structure, they lack detail clarity. Our method generates highly sharp and realistic images by reconstructing high-frequency details like road contours and ground textures

| Method | ThermoScenes | | | TI-NSD | | |
|---|---|---|---|---|---|---|
| | PSNR↑ | SSIM↑ | LPIPS↓ | PSNR↑ | SSIM↑ | LPIPS↓ |
| WaveGS ($s = 1$) | 29.507 | 0.956 | 0.264 | 32.080 | 0.927 | 0.194 |
| WaveGS ($s = 2$) | 32.849 | 0.950 | 0.196 | 33.294 | 0.946 | 0.200 |
| WaveGS ($s = 3$) | **33.671** | **0.978** | **0.064** | **34.952** | **0.958** | **0.182** |

Table 3: Ablation study of WaveGS with different wavelet scales ($s$) on the ThermoScenes and TI-NSD datasets. The best results for each metric are highlighted in **bold**.

However, it also produces a poor LPIPS score, suggesting potential perceptual artifacts.

**Analysis on Multi-scale Wavelet Decomposition** To investigate the influence of different wavelet decomposition levels on model performance, we conducted an ablation study evaluating WaveGS with varying decomposition scales ($s$). The model was trained and evaluated with configurations of $s = 1$, $s = 2$, and $s = 3$ on the ThermoScenes and TI-NSD benchmarks. As presented in Table 4, the quantitative results reveal a clear and compelling trend that differs from findings in prior work like WaveNeRF. For WaveGS, performance consistently and significantly improves as the decomposition scale increases from 1 to 3 across both benchmarks. This outcome suggests that for the WaveGS architecture, a hierarchical decomposition yields a powerful scene representation.

**Analysis on Loss Components.** We conducted an ablation study to evaluate the contribution of each proposed loss component on ThermoScenes. The quantitative results are shown in Table 2. Adding the wavelet sparsity loss $\mathcal{L}_{\text{sparsity}}$ leads to a clear improvement over the baseline. This confirms its effectiveness in encouraging a compact and structurally meaningful high-frequency representation. The resulting sparse activation map serves as an important prior for subsequent geometric refinement.We also evaluated the two deformation-related losses independently. The locality loss $\mathcal{L}_{\text{deform}}$ improves accuracy by focusing geometric adjustments on salient regions. The smoothness loss $\mathcal{L}_{\text{smooth}}$ enhances coherence by regularizing the deformation field. Finally, the complete model, which combines the three components, achieves the best performance on all metrics. This shows that sparsity and geometric regularization work well together to produce high-quality reconstructions.

## 5 CONCLUSION

In this paper, we propose WaveGS , which significantly improves the performance of novel view synthesis in thermal infrared imaging by leveraging the low-pass characteristic of heat conduction in the frequency domain. We parameterize the 3D scene using learnable wavelet coefficients, first optimizing the low-frequency components to capture the coarse scene structure, and progressively activating high-frequency coefficients to refine the details. Extensive experiments conducted on four datasets demonstrate that our method significantly outperforms state-of-the-art approaches.

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

## A    IMPLEMENTATION DETAILS

We implemented our WaveGS method in PyTorch, building upon the publicly available Scaffold-GS codebase. Our framework supports multiple data formats. For real-world datasets like TI-NSD, ThermoScenes and RGBT-Scenes, we adhere to the standard COLMAP data format. For the synthetic NTR dataset, we follow the Blender data format protocol.

Our scene initialization strategy is dataset-dependent. For all COLMAP-based datasets, we initialize the 3D Gaussians from the point cloud generated by Structure-from-Motion (SfM). On the synthetic NTR dataset, our method and other baselines start from a random point cloud initialization. In contrast, NTR-Gaussian is unique as it leverages the provided mesh for initialization on this specific dataset. This dependency on pre-processed data also explains why its results were not reproduced on other benchmarks; its open-source code requires a specific space features file but does not provide the means to generate it.

Regarding our model's specific parameters, we initialize the low-frequency approximation coefficients (LL sub-band) from a random distribution. All high-frequency detail coefficients (LH, HL, and HH) are initialized to zero. This encourages the model to first establish a coarse scene structure. We trained all models for 30,000 iterations on a single NVIDIA RTX 4090 GPU, using the Biorthogonal 6.8 (Bior6.8) wavele. A single Adam optimizer was used, but with distinct learning rates for the Gaussian attributes and the learnable wavelet coefficients.

## B    DATASET

Our model's effectiveness is validated across four distinct benchmarks for thermal novel view synthesis. A detailed overview is provided below.

- **RGBT-Scenes**: The RGBT-Scenes dataset is a real-world dataset designed for thermal 3D reconstruction and novel-view synthesis, consisting of aligned RGB and thermal images captured from multiple viewpoints across 10 different scenes using the commercial-grade handheld thermal-infrared camera FLIR E6 PRO (Teledyne FLIR), which can simultaneously capture RGB and thermal images . The camera has a resolution of 240×180, a field of view of 33°×25°, a temperature range from -20°C to 550°C, and a temperature accuracy of ±2% of the reading . The dataset includes over 1,000 RGB and thermal images, covering indoor and outdoor environments, various object sizes (from large structures to everyday items), different temperature variations (ranging from a 300°C difference to a 4°C difference), and both 360-degree and forward-facing scenarios, along with raw thermal camera images, MSX images, and camera pose data . Compared to existing datasets like Thermal-NeRF and ThermoNeRF, it includes both RGB and thermal images, applies multimodal calibration methods to align these images, ensures consistent thermal measurements across views, encompasses richer scene content, and makes calibration images available.

- **TI-NSD**: The TI-NSD dataset has several practical applications. In autonomous driving, it can help train models to better recognize objects in adverse weather conditions like fog or at night, as thermal infrared imaging isn't hindered by low light. For security surveillance, it enables the detection of intruders or abnormal activities in all - weather scenarios, as thermal signatures can reveal human presence even in darkness. In urban planning, the dataset can be used to analyze heat distribution in different areas, which is crucial for energy - efficient building design and understanding how different materials and structures interact with heat. Additionally, in remote sensing for environmental monitoring, TI-NSD can assist in identifying thermal anomalies in forests (such as early signs of wildfires) or water bodies (like thermal pollution), taking advantage of the all - weather and penetration capabilities of thermal infrared imaging.

- **ThermoScenes**: The ThermoScenes dataset is the first benchmark featuring paired RGB and thermal images for 3D scene reconstruction and novel view synthesis, comprising 16 diverse scenes—8 building facades (e.g., seasonal buildings, dorms) and 8 everyday objects (e.g., heated cups, laptops). Captured using a FLIR One Pro LT dual camera (thermal range: -20°C to 120°C, accuracy ±3°C), it provides aligned RGB-thermal image pairs with precise camera poses. Each scene includes training and test views (e.g., 107 train/15 test

for "Building (Spring)"), covering a wide temperature range from -16.2°C to 87.3°C. Raw thermal data (extracted from MSX images) is provided to avoid texture interference, making it ideal for evaluating multimodal methods like ThermoNeRF. It supports research in building energy analysis, non-destructive testing, and infrastructure inspection, with public access and plans for expansion.

- **NTR Dataset**: The NTR dataset is a dedicated benchmark for nighttime dynamic thermal reconstruction tasks. It encompasses four distinct outdoor scenes, including two urban scenes (S1, S2) mainly consisting of buildings and roads, and two suburban scenes (S3, S4) dominated by farmland and ponds . For each scene, aerial thermal infrared (TIR) images are captured at four time intervals during the night using a DJI Matrice 300 RTK drone equipped with a DJI H20T thermal infrared camera, which has a resolution of 640×512 and a temperature measurement range from -40°C to 550°C . Additionally, based on the high-precision 3D texture model from the UAV4DL dataset, the dataset provides accurately calibrated poses for all TIR images through a render-to-match framework and generates corresponding synthetic RGB images, effectively capturing the temporal variations in thermal radiation of objects under nighttime conditions to support research on dynamic thermal 3D reconstruction and novel viewpoint TIR image synthesis.

## C  HEAT CONDUCTION

**Heat Conduction.** The physical process of heat conduction is governed by the heat equation, a partial differential equation (PDE) that describes how the distribution of heat evolves over time in a given region. The fundamental principle is that heat flows from warmer to cooler areas, causing temperature gradients to smooth out.

Let $u(\mathbf{x}, t)$ represent the temperature at a spatial location $\mathbf{x} \in \mathbb{R}^n$ and time $t$. The heat equation is formulated as:

$$\frac{\partial u}{\partial t} = k\nabla^2 u \tag{17}$$

where $k$ is the thermal diffusivity, a material-specific property that determines the rate of heat transfer. The Laplacian operator, $\nabla^2 = \sum_{i=1}^{n} \frac{\partial^2}{\partial x_i^2}$, measures the local curvature of the temperature field. For a 2D case, where $\mathbf{x} = (x, y)$, this expands to:

$$\frac{\partial u}{\partial t} = k\left(\frac{\partial^2 u}{\partial x^2} + \frac{\partial^2 u}{\partial y^2}\right) \tag{18}$$

with an initial temperature distribution $u(x, y, 0) = f(x, y)$.

Solving this PDE directly can be complex. The Fourier Transform provides an elegant solution by converting the PDE in the spatial domain into a simpler ordinary differential equation (ODE) in the frequency domain. Let $\overline{u}(\boldsymbol{\omega}, t) = \mathcal{F}(u(\mathbf{x}, t))$ be the Fourier transform of the temperature field, where $\boldsymbol{\omega}$ is the frequency vector. Key properties of the Fourier transform are $\mathcal{F}(\partial_t u) = \partial_t \overline{u}$ and $\mathcal{F}(\nabla^2 u) = -\|\boldsymbol{\omega}\|^2 \overline{u}$. Applying these to the heat equation yields:

$$\frac{d\overline{u}(\boldsymbol{\omega}, t)}{dt} = -k\|\boldsymbol{\omega}\|^2 \overline{u}(\boldsymbol{\omega}, t) \tag{19}$$

This is a first-order ODE whose solution, given the transformed initial condition $\overline{f}(\boldsymbol{\omega}) = \mathcal{F}(f(\mathbf{x}))$, is:

$$\overline{u}(\boldsymbol{\omega}, t) = \overline{f}(\boldsymbol{\omega})e^{-k\|\boldsymbol{\omega}\|^2 t} \tag{20}$$

This solution holds for any number of dimensions. For the 3D case relevant to volumetric rendering, where $\boldsymbol{\omega} = (\omega_x, \omega_y, \omega_z)$, the solution is:

$$\overline{u}(\omega_x, \omega_y, \omega_z, t) = \overline{f}(\omega_x, \omega_y, \omega_z)e^{-k(\omega_x^2 + \omega_y^2 + \omega_z^2)t}. \tag{21}$$

The term $e^{-k\|\boldsymbol{\omega}\|^2 t}$ acts as a Gaussian low-pass filter. The magnitude of the frequency vector, $\|\boldsymbol{\omega}\|$, is high for sharp details and low for smooth variations. The equation shows that high-frequency components are attenuated exponentially faster over time $t$ than low-frequency components. This mathematical result perfectly mirrors the physical reality: sharp temperature differences (e.g., a hot spot on a cool surface) dissipate quickly, while large-scale, smooth temperature fields persist much longer. This physical prior is a core motivation for our work.

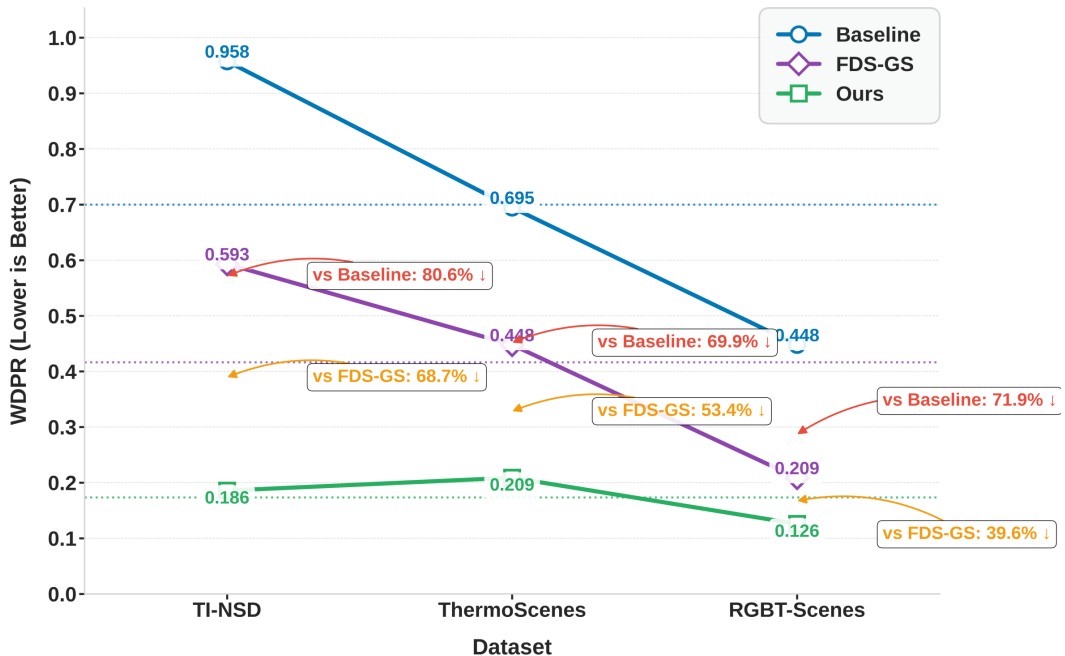

Figure 6: **Evaluation on high-frequency detail preservation.** We compare our method against Baseline and FDS-GS using the WDPR metric (lower is better), which quantifies the error in high-frequency detail reconstruction.

## D  ANALYSIS ON HIGH-FREQUENCY DETAIL PRESERVATION.

To further assess frequency-specific reconstruction quality, we introduce the *Wavelet Decomposition Power Ratio* (WDPR) Sun et al. (2024), which quantifies the discrepancy in high-frequency components. Specifically, we perform a $\lambda$-level wavelet decomposition as:

$$\text{WDPR}(y_{\text{true}}, y_{\text{syn}}, \lambda) = \frac{|P\left(W(y_{\text{true}}, \lambda)\right) - P\left(W(y_{\text{syn}}, \lambda)\right)|}{P\left(W(y_{\text{true}}, \lambda)\right)} \tag{22}$$

where $W(\cdot, \lambda)$ denotes the wavelet coefficients at level $\lambda$, and $P(\cdot)$ computes the corresponding signal power. The quantitative results, summarized in Figure 6, validate the superiority of our method in preserving high-frequency details. According to the WDPR metric (lower is better), our approach consistently and significantly outperforms both the Baseline and the enhanced FDS-GS method across all datasets. The advantage is particularly pronounced on the TI-NSD dataset, where our model reduces the high-frequency reconstruction error by a remarkable 80.6% against the baseline and 68.7% against FDS-GS. This substantial margin of improvement is maintained across the ThermoScenes and RGBT-Scenes datasets, confirming the robustness and generalization capability of our approach. By explicitly modeling the sparse and nonuniform nature of high-frequency information, our method reconstructs fine-grained textures and sharp edges.

## E  ANALYSIS ON WAVELET-DOMAIN ENERGY CHARACTERISTICS

To validate the effectiveness of our proposed wavelet-decomposition field and high-frequency guided deformation, we first analyze the intrinsic properties of thermal signals and then present a quantitative comparison of reconstruction fidelity. As illustrated in Figure 7, our analysis of signal energy distribution in the wavelet domain provides the theoretical foundation for our methodology. The left panel reveals that signal energy is predominantly concentrated in the low-frequency subband, while high-frequency components exhibit natural sparsity, with their energy being orders of magnitude lower. This inherent sparsity justifies our strategy of representing features in the wavelet domain and applying L1 regularization to enforce a compact and efficient representation. Furthermore, the right panel demonstrates that the high-frequency energy distribution varies

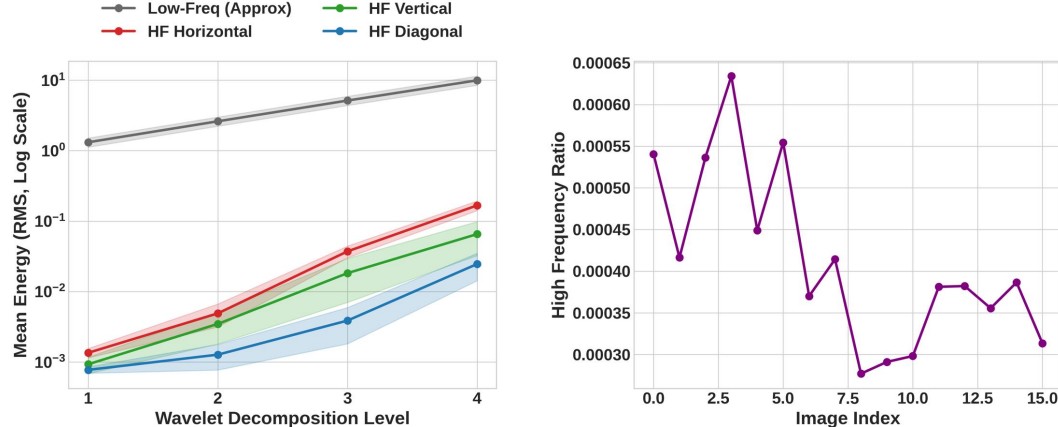

Figure 7: **Wavelet-Domain Energy Characteristics of Thermal Imagery.** (Left) A log-scale plot showing that signal energy is predominantly concentrated in the low-frequency components, while high-frequency (HF) components are sparse. (Right) The proportion of high-frequency energy is highly variable across different scenes, indicating that structural details are not uniformly present.

| Method | ThermoScenes | | | TI-NSD | | |
|---|---|---|---|---|---|---|
| | PSNR↑ | SSIM↑ | LPIPS↓ | PSNR↑ | SSIM↑ | LPIPS↓ |
| WaveGS ($s = 1$) | 29.507 | 0.956 | 0.264 | 32.080 | 0.927 | 0.194 |
| WaveGS ($s = 2$) | 32.849 | 0.950 | 0.196 | 33.294 | 0.946 | 0.200 |
| WaveGS ($s = 3$) | **33.671** | **0.978** | **0.064** | **34.952** | **0.958** | **0.182** |

Table 4: Ablation study of WaveGS with different wavelet scales ($s$) on the ThermoScenes and TI-NSD datasets. The best results for each metric are highlighted in **bold**.

markedly across scenes, highlighting the non-uniform spatial distribution of geometric details. This observation underscores the necessity of an adaptive mechanism that targets structurally critical regions. Our high-frequency guided deformable field addresses this requirement by employing a learned mask to apply geometric corrections exclusively in relevant areas.

## F ANALYSIS ON MULTI-SCALE WAVELET DECOMPOSITION

The superior performance can be attributed to a fundamental distinction in methodology. WaveGS builds its 3D feature field upon a **grid-free, anchor-based Vector-Matrix (VM) decomposition**, parameterizing the factors with learnable wavelet coefficients. This design circumvents the challenges of feature map resolution and padding at higher decomposition levels, which are often encountered by methods that rely on explicit cost volumes and plane sweeps. A greater number of scales ($s = 3$) allows for a more refined disentanglement of the scene into a richer combination of low- and high-frequency components, a process visually conceptualized in **Figure 8**. This enables the model to capture macroscopic thermodynamic distributions with smoother basis functions (analogous to the LL sub-bands) while simultaneously representing fine-grained geometric edges and structural details with a more diverse set of high-frequency bases (the LH, HL, and HH sub-bands). Furthermore, a deeper decomposition provides a more flexible framework for embedding the low-pass physical prior of heat conduction. The model can more accurately concentrate energy in the low-frequency bands while performing sparse and precise detail refinement in the high-frequency subbands, a characteristic quantitatively visualized by the sparse, high-magnitude coefficients in **Figure 9**, leading to superior reconstruction fidelity. In conclusion, the experimental results demonstrate that increasing the wavelet decomposition scale significantly enhances the performance of WaveGS. This is primarily due to its efficient grid-free, anchor-based architecture, which effectively capitalizes on the benefits of hierarchical feature disentanglement without incurring the computational and memory bottlenecks associated with volume-based methods at higher scales.

# G ADDITIONAL VISUALIZATIONS

To further demonstrate the superior performance of our proposed WaveGS, we conducted extensive qualitative evaluations. We compared its novel view synthesis results against current state-of-the-art methods on several benchmark datasets. These supplementary results clearly highlight the advantages of our approach. It generates views with richer details and fewer artifacts. This is particularly evident in regions with complex thermal patterns and fine structures. This improvement is a direct result of our physics-informed wavelet splatting mechanism. This mechanism more effectively preserves high-frequency details within the scene.

**TI-NSD** Figure 10 presents the qualitative comparison on the TI-NSD dataset. The results show that our method achieves the highest clarity and detail fidelity. This applies to the fine structures of windows, textures on building roofs, and outlines of small objects. The synthesized images from our method are closest to the Ground Truth. In contrast, other methods like Thermal3D-GS, ThermalGaussian, Scaffold-GS, and FDS-GS exhibit various issues. These issues include blurring, loss of detail, or shape distortion when rendering these complex structures. This demonstrates that WaveGS better preserves thermal structures and detail sharpness.

**RGBT-Scenes** Figure 11 provides a qualitative comparison on the RGBT-Scenes dataset. This dataset features diverse and challenging scenes. Examples include nighttime lights, complex vegetation, and man-made objects with fine structures like scooters. WaveGS performs exceptionally well in these scenarios. Its results are highly consistent with the Ground Truth in terms of detail. For instance, in the third row, our method clearly reproduces the thermal patterns among the leaves of the trees. Other methods show significant blurring and artifacts. Similarly, in the fourth row, WaveGS accurately renders the scooter's contour and temperature distribution. This showcases its powerful capability for detail capture.

**ThermoScenes** The comparison on the ThermoScenes dataset is shown in Figure 12. This figure further validates the superiority of WaveGS. It excels at generating sharp structural contours and accurate thermal regions. As highlighted in the red boxes, our method produces sharper outlines for building edges and roof details. It also renders more precise thermal patterns. The results are clearly superior to baseline methods such as Thermal3DGS, ThermalGaussian, Scaffold-GS, and FDS-GS.

**NTR Dataset** Finally, we conducted evaluations on the NTR dataset. The results are presented in Figure 13 and Figure 14. In the s2 scene (Figure 13), the aerial view generated by our method (Ours) is significantly better. It shows greater clarity in the road network and fewer artifacts across the entire scene compared to 2DGS and other methods. Our result most closely resembles the Ground Truth (GT). Similar observations were made in the s4 scene (Figure 14). WaveGS maintains high structural fidelity and detail clarity when reconstructing large-scale urban scenes, whereas other methods suffer from noticeable blurring and distortion.

In summary, these extensive qualitative comparisons collectively prove the advanced performance of WaveGS for thermal novel view synthesis. Our method effectively utilizes wavelet transform to represent and render high-frequency thermal information. As a result, it achieves precise reconstruction of fine structures and thermal patterns across a wide variety of complex scenes.

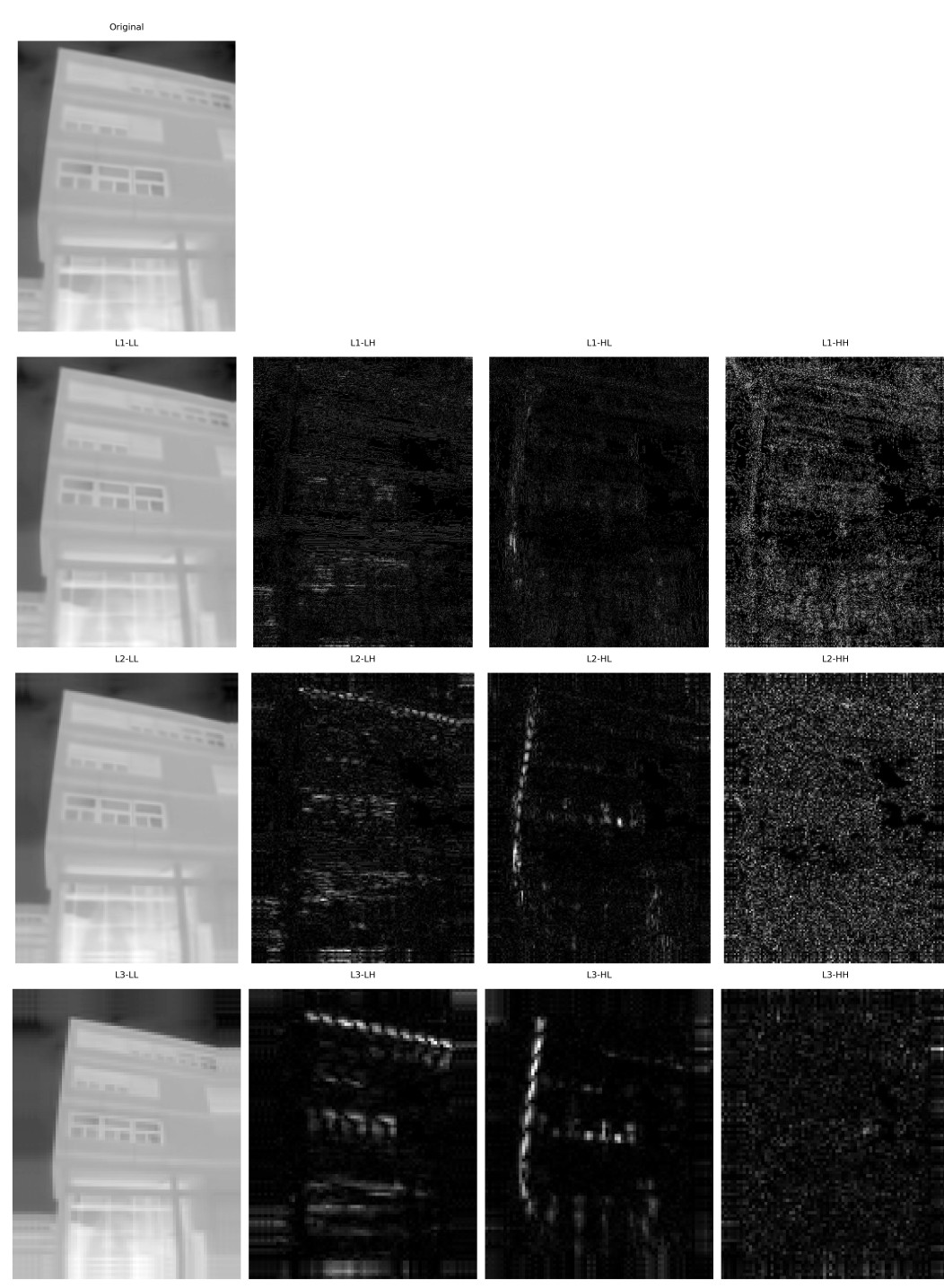

Figure 8: **Visualization of a 3-level 2D Discrete Wavelet Transform (DWT).** The original image (top left) is progressively decomposed into an approximation sub-band (LL) and three detail sub-bands: horizontal (LH), vertical (HL), and diagonal (HH). The LL sub-band retains the low-frequency, coarse representation of the image, while the detail sub-bands effectively isolate the directional edge features at each respective scale.

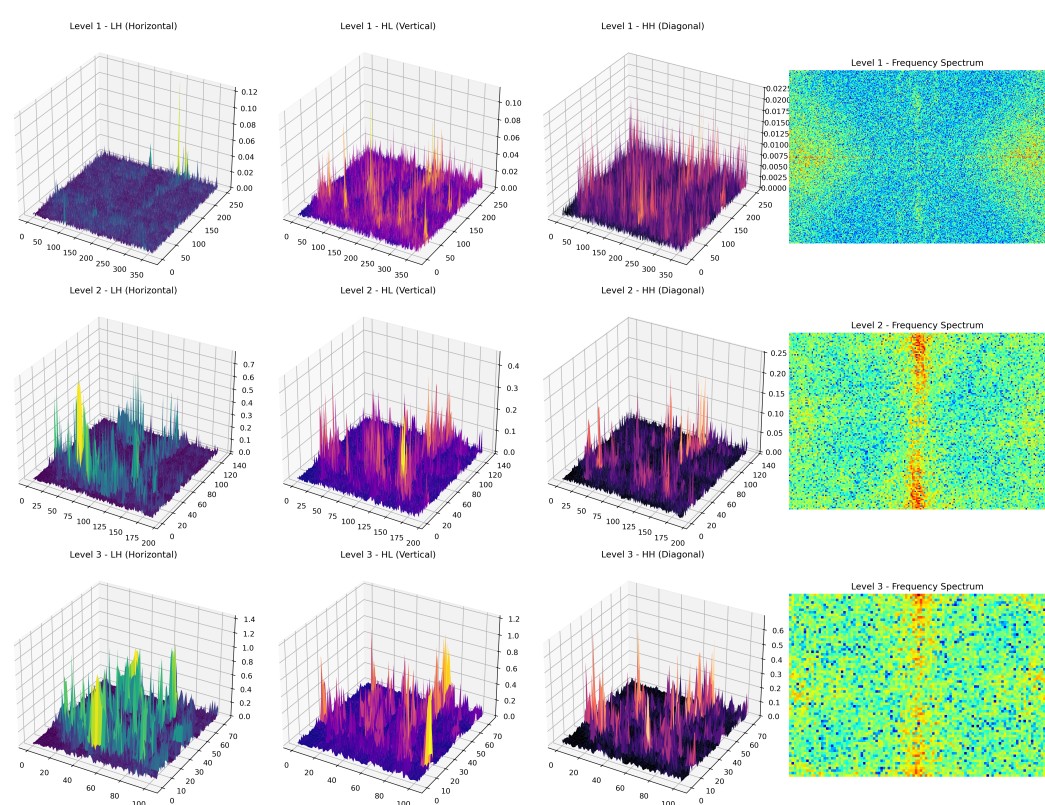

Figure 9: **Quantitative and frequency-domain analysis of DWT detail coefficients.** The 3D surface plots visualize the magnitude of coefficients in the detail sub-bands (LH, HL, HH), where peaks indicate strong edge features. The frequency spectrum plots on the right reveal the energy distribution at each decomposition level. Notably, the prominent vertical line in the Level 2 spectrum confirms a strong dominance of horizontal features, corresponding to the building's structural lines.

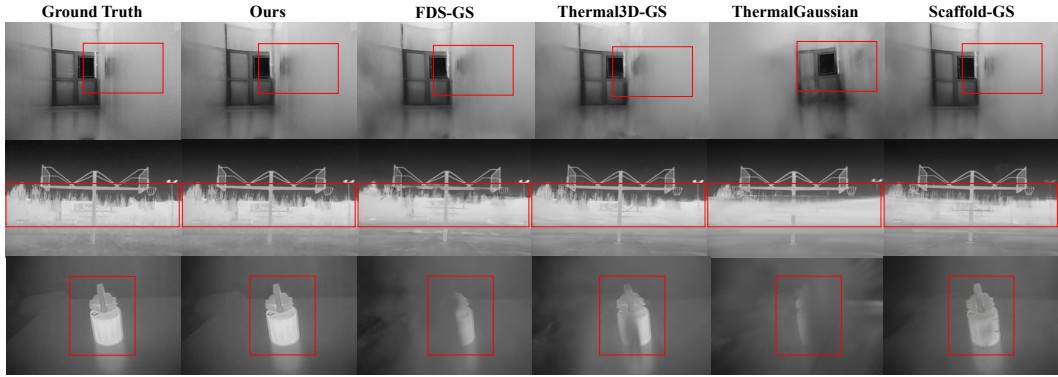

Figure 10: **Qualitative comparison on TI-NSD** From left to right: Ground Truth, Our Method, Thermal3DGS, Thermal-Gaussian, Scaffold-GS and FDS-GS. Our method better preserves thermal structures and detail sharpness.

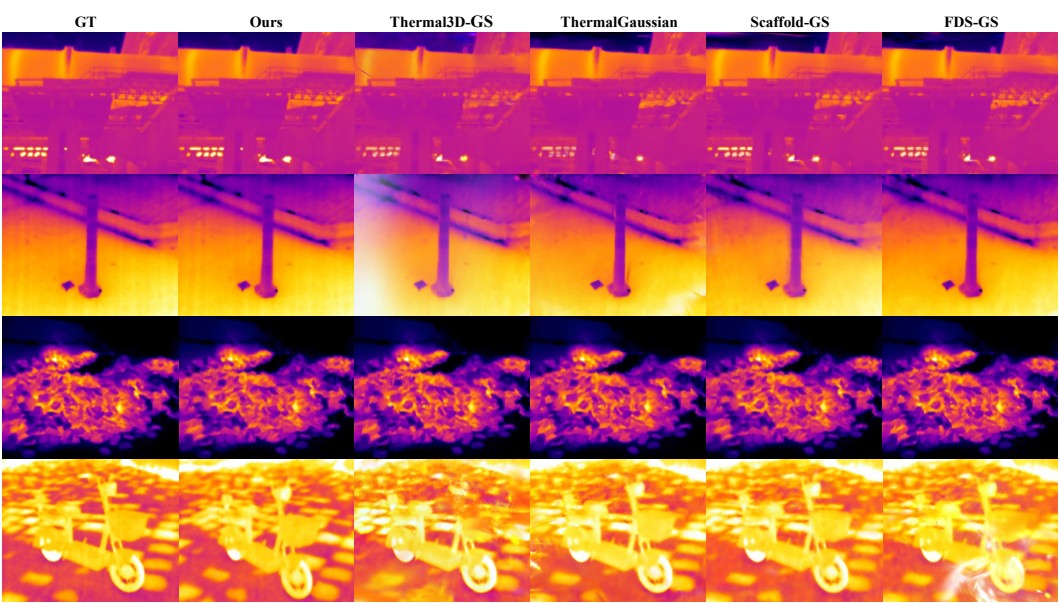

Figure 11: **Qualitative comparison on RGBT-Scenes.** From left to right: Ground Truth, Our Method, Thermal3DGS, Thermal-Gaussian, Scaffold-GS and FDS-GS. Our method better preserves thermal structures and detail sharpness.

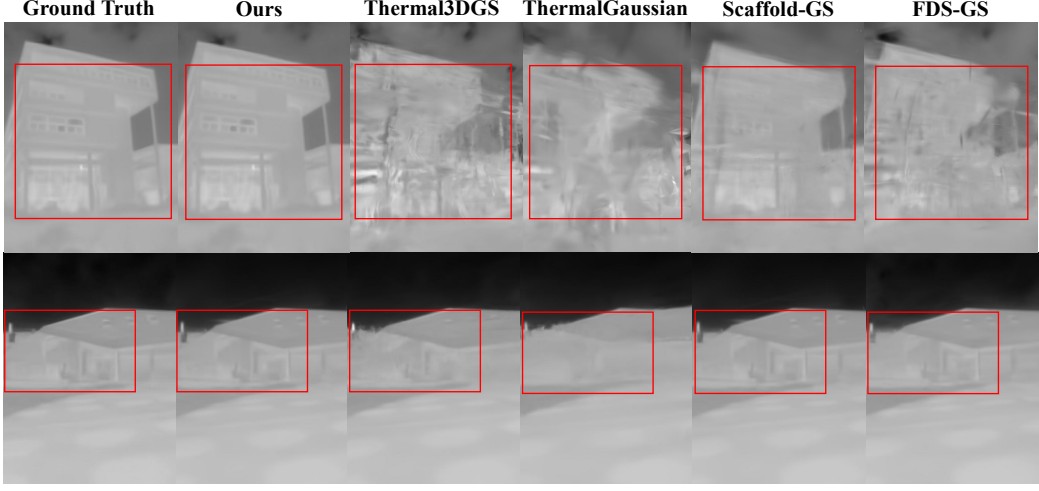

Figure 12: **Qualitative comparison on ThermoScenes.** WaveGS generates sharper structural contours and more accurate thermal regions than baseline methods.

| GT | Ours | 2DGS | NTR-Gaussian | Thermal3DGS | ThermalGaussian |

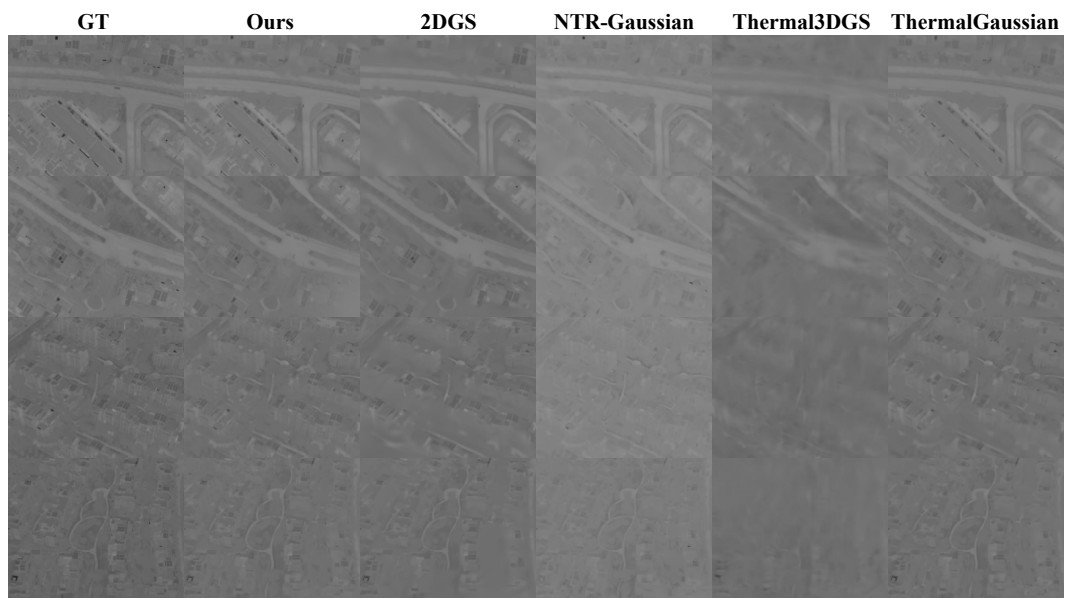

Figure 13: **Qualitative comparison on NTR (s2).**

| GT | Ours | 2DGS | NTR-Gaussian | Thermal3DGS | ThermalGaussian |

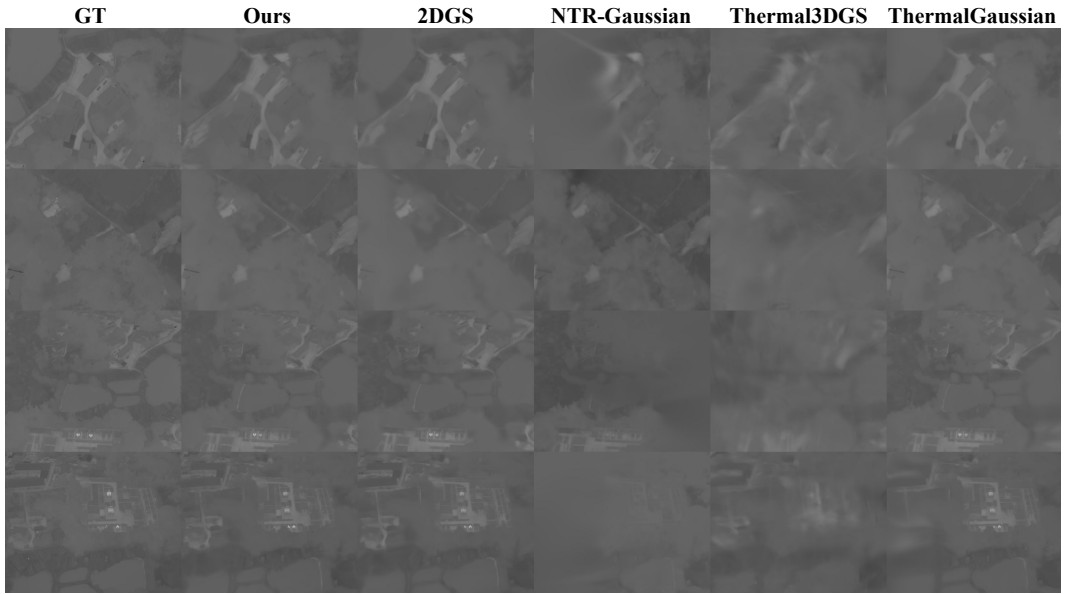

Figure 14: **Qualitative comparison on NTR (s4).**

## H   LLMs USAGE

We utilized Large Language Models (LLMs) solely for language editing and refinement. Their application was limited to correcting grammar, optimizing sentence structure, and enhancing the overall readability of the manuscript. All core research contributions, including the conceptual framework, methodology, data analysis, and conclusions, are entirely the original work of the authors.

