# OpenReview forum: "WaveGS: Physics-Inspired Wavelet Splatting for Thermal Novel View Synthesis"
_ICLR.cc/2026/Conference — ICLR 2026 Conference Withdrawn Submission_

### Official Review · Reviewer_uXUB · 2025-10-27

**Soundness:** 2
**Presentation:** 3
**Contribution:** 2
**Rating:** 4
**Confidence:** 3

**Summary:**

The paper proposes WaveGS, a physics-inspired neural 3D scene representation for thermal infrared novel view synthesis. The key motivation is that thermal radiation obeys heat diffusion, which behaves like a low-pass filter in the frequency domain: most energy concentrates in low-frequency components while high-frequency details are sparse and often noisy. WaveGS models a 3D thermal scene by (1) representing the scene as a continuous Vector-Matrix (VM) decomposition anchored on a sparse set of Scaffold-GS-like anchor points instead of dense voxel grids, enabling efficient querying and real-time Gaussian splatting. (2) Parameterizing both 1D vector factors and 2D matrix factors directly in the wavelet domain, explicitly separating low-frequency “smooth thermal field” components from multi-scale high-frequency subbands. (3) Learning a sparsity-inducing high-frequency mask that (i) suppresses noisy or spurious high-frequency coefficients via L1-style regularization, and (ii) guides a localized anchor deformation field which refines geometry only where structurally meaningful high-frequency responses exist, while enforcing smooth and spatially local corrections. The masked and modulated wavelet coefficients are then reconstructed back into the spatial domain via a differentiable inverse wavelet transform, yielding per-anchor features that drive Gaussian rendering. Experiments across four thermal datasets (RGBT-Scenes, ThermoScenes, TI-NSD, and NTR) show that WaveGS outperforms baselines including generic 3D Gaussian Splatting variants, frequency-aware methods such as FDS-GS, and physics-motivated thermal renderers such as Thermal3D-GS, ThermalGaussian, and NTR-Gaussian, often achieving the best PSNR/SSIM/LPIPS while maintaining real-time rendering (>200 FPS).

**Strengths:**

- Strong physical motivation. The paper grounds its design in the physics of heat conduction (the heat equation), which implies exponential attenuation of high spatial frequencies in the Fourier domain. This provides a principled reason to bias the model toward low-frequency structure and sparse high-frequency activations, rather than just adding ad-hoc frequency penalties.
- Wavelet-domain parameterization of a 3D thermal field. Instead of learning everything in the spatial domain, WaveGS learns multi-scale wavelet coefficients (LL / LH / HL / HH) for both 1D vector and 2D matrix VM factors, then reconstructs them via differentiable inverse wavelet transform. This is claimed to be the first such parameterization specifically tailored to thermal novel view synthesis.

**Weaknesses:**

- Novelty vs. system integration.
While the physics motivation and the high-frequency-driven deformation are compelling, a large portion of the pipeline is a careful fusion of existing components: Scaffold-GS-style anchors and Gaussian rendering, VM/TensoRF-like low-rank factorization for continuous fields, multi-scale wavelets for frequency localization, sparsity regularization, etc. The paper could do more to clearly separate what is genuinely new from what is reused. Right now, I feel the method is a sophisticated “all-star mashup,” and it is not yet quite clear which pieces are indispensable.
- Ablation scope.
The ablations are informative but mostly internal. We do not see an external control such as:
(i) Scaffold-GS + learned local deformation but without wavelet-domain modeling (i.e., everything in spatial domain).
(ii) FDS-GS + the proposed high-frequency sparsity mask and deformation field.
Such comparisons would better validate the claim that representing the scene in the wavelet domain is the main driver of improvement (and not simply the presence of a localized deformation MLP on top of an already strong Gaussian baseline).
- No video / multiview flythrough evidence.
The paper mainly shows per-view still images and quantitative metrics (PSNR/SSIM/LPIPS), but does not convincingly present continuous camera flythroughs or multi-view trajectories (e.g., a smooth orbit around the scene, or UAV-style forward motion) demonstrating temporal / geometric consistency across frames.

**Questions:**

These are questions where the author's rebuttal could meaningfully change my rating.
- Modularity of contributions.
Can you provide an ablation where you only add the wavelet-domain parameterization (multi-scale low/high-frequency coefficients) on top of a standard Gaussian Splatting baseline without the deformation field and sparsity mask? Conversely, can you show Scaffold-GS + localized deformation (guided by image-space edges or some heuristic) but still trained purely in the spatial domain, no wavelet? This would help isolate whether the true gain comes from the physics-inspired frequency modeling vs. the geometric refinement machinery.
- Deformation stability.
The deformation field is regularized both for locality (penalizing displacement where the high-frequency mask is low) and for smoothness across neighboring anchors. Could you report whether this deformation ever “runs away,” e.g., causing anchors to drift and distort global geometry? Are there practical bounds or clamping heuristics you apply during training?
- Video evidence of 3D consistency.
Can you share multi-frame novel-view renderings (e.g., a smooth camera sweep, orbit, or UAV-style forward motion) for at least one of the thermal sequences if it is possible? This would directly address the concern that your method might be fitting each training view independently, rather than learning a stable, viewpoint-invariant 3D thermal field. A short flythrough video is often the most convincing qualitative proof of geometric consistency in Gaussian splatting / NeRF-style work. But if the rebuttal does not allow to include videos, this will not lead to a deduction of my rating.

---

### Official Review · Reviewer_rjCo · 2025-10-29

**Soundness:** 3
**Presentation:** 2
**Contribution:** 2
**Rating:** 4
**Confidence:** 4

**Summary:**

The paper proposes WaveGS, a physics-inspired framework for 3D thermal infrared novel view synthesis. The key idea is to leverage the low-pass characteristic of heat conduction and represent the 3D thermal field directly in the wavelet frequency domain. Specifically, the authors decompose the 3D scene features into low-frequency (smooth thermal distributions) and high-frequency (structural details) components using a learnable wavelet basis. A high-frequency mask suppresses noise while preserving salient details and guides a geometric deformation field that locally adjusts anchor positions. The modulated wavelet coefficients are reconstructed through a differentiable inverse wavelet transform to produce the final feature field.

The framework is evaluated on four datasets, including TI-NSD, RGBT-Scenes, ThermoScenes, and NTR, showing consistent improvements over existing baselines such as Thermal3D-GS, ThermalGaussian, Scaffold-GS, and FDS-GS in terms of PSNR, SSIM, and LPIPS.

**Strengths:**

1. The method is well-motivated and clearly articulates the connection between thermal physics and representation learning. The use of the heat equation’s low-pass property to constrain 3D representation learning is conceptually reasonable and physically interpretable.

2. The integration of wavelet decomposition into 3D Gaussian Splatting (3DGS) via a Vector-Matrix decomposition is original. It combines physics priors (heat diffusion) with frequency-domain learning, representing an extension of Gaussian splatting methods into thermal domains.

3. Experiments on four diverse datasets comprehensively demonstrate the superiority of the proposed approach. The results show that WaveGS yields sharper, more faithful thermal reconstructions and preserves structural fidelity better than previous physics-based or frequency-guided methods.

4. The work advances thermal infrared view synthesis, a challenging but practically important domain for robotics, inspection, and medical applications. The physics-guided approach provides a blueprint for integrating domain-specific priors into neural rendering frameworks.

**Weaknesses:**

1. A large proportion of the paper describes pre-existing methods (e.g., Scaffold-GS, Vector-Matrix decomposition, deformation fields), making it difficult to isolate which specific components constitute the novel contribution. A clearer demarcation of new versus inherited components would help.

2. The paper does not explicitly discuss Wavelet-GS or other recent works that introduce wavelet transforms into Gaussian Splatting. Comparing with these would clarify the unique contribution of WaveGS beyond general frequency modeling.

3. While the ablation study shows significant gains when regularization losses (Lsparsity, Ldeform, Lsmooth) are added, it remains unclear how much improvement stems from the wavelet modeling itself versus from better regularization strategies. A more granular ablation isolating these effects would be valuable.

4. The paper reports only accuracy-based metrics (PSNR, SSIM, LPIPS). Including training time, model size, and rendering speed would strengthen the evaluation and demonstrate the method’s practicality for real-time or large-scale applications.

**Questions:**

1. Can the authors provide efficiency metrics of both the propsed method and the compared methods?

2. How does WaveGS differ from Wavelet-GS, or other frequency-aware representations? Were these baselines considered or excluded for specific reasons?

---

### Official Review · Reviewer_qRbS · 2025-10-29

**Soundness:** 3
**Presentation:** 2
**Contribution:** 2
**Rating:** 6
**Confidence:** 2

**Summary:**

This paper introduces WaveGS, a physics-inspired framework for thermal infrared novel view synthesis. The key idea is to exploit the low-pass characteristics of heat conduction by modeling thermal scene representations in the wavelet (frequency) domain. The method employs a Vector-Matrix decomposition parameterized by learnable wavelet bases, separating the thermal field into low- and high-frequency components. A sparsity-inducing high-frequency mask suppresses infrared noise while guiding a learnable geometric deformation field to refine local geometry. The approach integrates these components via a differentiable inverse wavelet transform, allowing end-to-end optimization. Experiments across four datasets (RGBT-Scenes, ThermoScenes, TI-NSD, and NTR) demonstrate superior PSNR, SSIM, and LPIPS compared to both frequency-based (FDS-GS) and physics-guided baselines (Thermal3D-GS, ThermalGaussian).

**Strengths:**

1. **Novel and Physically Grounded Approach**: The core contribution—using the physical property of heat conduction (the low-pass filtering effect) to guide the representation via wavelet decomposition—is well-motivated. This domain-specific regularization is an elegant way to stabilize the thermal NVS problem.

2. **Effective Solution for Blurriness**: Current thermal NVS methods, particularly those based on NeRF or standard 3DGS, struggle with blurry thermal boundaries and ghosting due to the difficulty of optimizing highly textured, non-color data. WaveGS directly addresses this by separating the smooth LF background from the sharp HF details, leading to noticeably sharper structural contours in the qualitative results.

3. **Strong Empirical Results**: The method demonstrates significant quantitative improvements across key metrics (PSNR, SSIM, LPIPS) over strong state-of-the-art baselines like Thermal-3DGS.

**Weaknesses:**

1. **Overemphasis on low-pass priors**: While thermal imagery is dominated by low frequencies, some real-world settings (e.g., textured heated surfaces) may violate this assumption; the paper could discuss such limitations more explicitly.

2. **Lack of analysis on computational cost**: The added wavelet decomposition and inverse transform steps might introduce overhead. The Runtime analysis and comparisons with baselines are missing.

3. **Insufficient comparison to frequency-aware baselines**: Works like WaveNeRF and FreGS are only briefly mentioned in related work; it would help to see a direct quantitative comparison.

**Questions:**

1. What is the exact runtime and memory overhead compared to baselines during training and inference?

2. How sensitive is the method to major hyperparameters (e.g., the weighting of sparsity and deformation losses, the number of decomposition levels, or the mask threshold)?

---

### Official Review · Reviewer_ttGs · 2025-11-02

**Soundness:** 2
**Presentation:** 2
**Contribution:** 2
**Rating:** 2
**Confidence:** 2

**Summary:**

This paper tackles the problem of thermal novel view synthesis, where the input is infrared images, and the output is a 3D thermal scene. This is harder than regular RGB scenes since thermal data lack texture and hence cause more ambiguities in geometry.

This model replaces gaussians with wavelet bases. Each anchor point is associated with a wavelet feature field so that low and high frequencies can be separated. The model also learns a mask for which high-frequency regions to keep. There is also a deformation field to adjust the anchor positions. Rendering is done with inverse wavelet transform, essentially splatting the wavelet bases instead of gaussians.

**Strengths:**

Interesting attack angle using GS. This is something that I was not expecting GS to be helpful for.

I like how physics is wired in during the design of this model. These are inductive biases that help solve domain-specific problems like this.

Good ablation studies that justify the choice of wavelet scales, losses, and bases.

**Weaknesses:**

Core method itself is basically Scaffold-GS, so a bit limited in that regard.

Unclear to me if the heavy physics/math actually helps the reader understand the approach. Especially I was expecting some inductive biases from thermodynamic simulation, etc., but looks like they are just a motivation for the low-pass prior?

**Questions:**

What do you use the novel views for? Maybe I am no expert in this field, but the applications mentioned by the paper don’t sound super compelling to me.

Why are wavelets preferred over fourier bases here?

Are there any physical interpretations that can be made out of the optimized GS field? It’d be cool to link some optimized results back to the physics.

---

### Note · Authors · 2025-12-10

I have read and agree with the venue's withdrawal policy on behalf of myself and my co-authors.